# Pre-Existing Tumoral B Cell Infiltration and Impaired Genome Maintenance Correlate with Response to Chemoradiotherapy in Locally Advanced Rectal Cancer

**DOI:** 10.3390/cancers12082227

**Published:** 2020-08-10

**Authors:** Juan M. Sendoya, Soledad Iseas, Mariana Coraglio, Mariano Golubicki, Juan Robbio, Ruben Salanova, Mirta Kujaruk, Vanesa Mikolaitis, Mariana Rizzolo, Gonzalo Ruiz, Ana Cabanne, Ubaldo Gualdrini, Guillermo Mendez, Stella Hirmas, Cecilia Rotondaro, Julieta Viglino, Martín Eleta, Elmer Fernandez, Martín Abba, Osvaldo Podhajcer, Enrique Roca, Andrea S Llera

**Affiliations:** 1Laboratorio de Terapia Molecular y Celular - Genocan, Fundación Instituto Leloir, Buenos Aires 1405, Argentina; jmsendoya@leloir.org.ar (J.M.S.); juliviglino@gmail.com (J.V.); opodhajcer@leloir.org.ar (O.P.); 2Consejo Nacional de Investigaciones Científicas y Técnicas (CONICET), Buenos Aires 1425, Argentina; crotondaro@leloir.org.ar (C.R.); efernandez@cidie.ucc.edu.ar (E.F.); mcabba@gmail.com (M.A.); 3Hospital de Gastroenterología Carlos Bonorino Udaondo, Buenos Aires 1264, Argentina; soledadiseas@gmail.com (S.I.); mcoraglio@hotmail.com (M.C.); mariano.golubicki@gmail.com (M.G.); rubensalanova@hotmail.com (R.S.); kujarukmirta@gmail.com (M.K.); vanesa@mikolaitis.com.ar (V.M.); mrizzolo@intramed.net (M.R.); gonzalogabrielruiz@yahoo.com.ar (G.R.); anacabanne@gmail.com (A.C.); ugualdrini@gmail.com (U.G.); mendezdoc@hotmail.com (G.M.); stellamh@gmail.com (S.H.); 4Intergrupo Argentino para el Tratamiento de los Tumores Gastrointestinales (IATTGI), Buenos Aires 1264, Argentina; robbiojuan@gmail.com (J.R.); enlroca@yahoo.com.ar (E.R.); 5Biomakers, Buenos Aires 1119, Argentina; 6Imaxe, Buenos Aires 1120, Argentina; martin.eleta@imaxe.com.ar; 7Centro de Investigación y Desarrollo en Inmunología y Enfermedades Infecciosas (CIDIE), Universidad Católica de Córdoba, Córdoba 5016, Argentina; 8Centro de Investigaciones Inmunológicas Básicas y Aplicadas (CINIBA), Facultad de Ciencias Médicas, Universidad Nacional de La Plata, La Plata 1900, Argentina

**Keywords:** rectal cancer, immune response, gene expression, neoadjuvant chemoradiotherapy, biomarker

## Abstract

Locally advanced rectal cancer (LARC) remains a medical challenge. Reliable biomarkers to predict which patients will significantly respond to neoadjuvant chemoradiotherapy (nCRT) have not been identified. We evaluated baseline genomic and transcriptomic features to detect differences that may help predict response to nCRT. Eligible LARC patients received nCRT (3D-LCRT 50.4 Gy plus capecitabine 825 mg/m${}^{2}$/bid), preceded by three cycles of CAPOX in high systemic-relapse risk tumors, and subsequent surgery. Frozen tumor biopsies at diagnosis were sequenced using a colorectal cancer panel. Transcriptomic data was used for pathway and cell deconvolution inferential algorithms, coupled with immunohistochemical validation. Clinical and molecular data were analyzed according to nCRT outcome. Pathways related to DNA repair and proliferation (*p* < 0.005), and co-occurrence of *RAS* and *TP53* mutations (*p* = 0.001) were associated with poor response. Enrichment of expression signatures related to enhanced immune response, particularly B cells and interferon signaling (*p* < 0.005), was detected in good responders. Immunohistochemical analysis of CD20${}^{+}$ cells validated the association of good response with B cell infiltration (*p* = 0.047). Findings indicate that the presence of B cells is associated with successful tumor regression following nCRT in LARC. The prevalence of simultaneous *RAS* and *TP53* mutations along with a proficient DNA repair system that may counteract chemoradio-induced DNA damage was associated with poor response.

## 1. Introduction

Over the last few decades, progress in the multimodal treatment of locally advanced rectal cancer (LARC) has evolved, substantially improving local disease control [1]. Preoperative chemoradiotherapy (nCRT) followed by total mesorectal excision (TME) and postoperative adjuvant treatment has been associated with considerable advantages over postoperative approaches, mainly offering improvement on the local recurrence rate [2]. Distant metastases remain the most frequent disease recurrence [3].

The current therapeutic standard for LARC treatment is total neoadjuvant therapy (TNT) which comprises induction fluorouracil- and oxaliplatin-based chemotherapy followed by CRT. TNT has been associated with more efficient delivery of systemic therapy and higher response rates [4,5,6]. Together, TNT and CRT are referred to as nCRT throughout this manuscript. On average, 20–30% of patients achieve a complete pathological response (pCR) after nCRT, while 50% of patients have a partial response and the remaining subgroup (20–30%) has minimal or no tumor regression [7,8]. The peak of tumor regression has been estimated to occur at least 8–12 weeks after treatment completion [9]. The quality of response to nCRT has been reported to be an independent predictor of either disease progression or overall survival [10,11]. We are currently lacking an accurate definition to assess response. In the seventh edition manual of the American Joint Committee on Cancer (AJCC), a pathological tumor regression (pTRG) score was introduced which has been shown to be superior to previously used systems [11,12,13]. Patients with pCR following nCRT (pTRG score of 0) have excellent long-term outcomes.Those with moderate, minimal, and no response have progressively worse outcomes [11]. Response assessment by MRI (mrTRG) has been proposed as a surrogate of pTRG [14], however retrospective studies have reported poor agreement between mrTRG and pTRG [15]. Lastly, the neoadjuvant rectal (NAR) score [16] has been proposed as a surrogate for prognosis but its prognostic value has been questioned [17].

Research to elicit the biological nCRT response determinants is a means of enabling risk stratification for better clinical decision-making. Different approaches to identify predictive molecular biomarkers have been published over the last decade [18,19,20,21,22,23,24,25,26], most solely based on molecular signatures derived from differential gene expression analyses. The lack of validation of such findings may be related to differences in patient selection, treatments and technologies, tumor heterogeneity, small sample sizes and, most importantly, varying definitions used for classifying tumor response. Few studies have performed in-depth analyses of the biological processes related to sensitivity or resistance to nCRT or integrated this with clinical data. We performed an integrative analysis of histological and molecular factors correlated with nCRT response in eligible LARC patients treated consecutively in a single institution. We used a comprehensive molecular approach (genome, transcriptome, protein levels) which includes the role of the tumor microenvironment. We report a robust collection of curated pathways and features at the mutational and transcriptomic levels that were found to be significantly associated with response to nCRT. Our results may lay the foundations for developing predictive biomarkers for LARC in the near future.

## 2. Results

### 2.1. Patient Population and Response Assessment

A total of 50 eligible LARC patients were enrolled in the study. Demographic, histopathological and clinical data are displayed in Table 1. Patients had a median age of 61 (range 34–75) years, and 37 were male (74%). In this series, 27 patients (54%) were allocated to TNT and the rest received CRT (see study diagram and molecular data analyzed in Figure 1). All patients were assessed for response no earlier than 8 weeks after completing radiotherapy. cCR was achieved by 3 patients (6%), 46 patients (92%) underwent TME surgery and 1 patient (2%) had local disease progression before surgery. Median time between the end of radiotherapy completion and surgery was 16 weeks. The TME R0 resection rate was 93% (44/47). In agreement with previous reports, almost all patients with mucinous histology responded poorly to nCRT treatment (7/8).

As of August 2019, with a median follow up of 25 months (range 13–45 months) since diagnosis, 19/50 patients had recurrent disease, with a pelvic recurrence in 6 patients (32%), distant metastases in 9 patients (47%), and both in 4 patients (21%). As a result of the disease, 11/19 (58%) had died. The remaining 31/50 patients were alive with no evidence of disease progression.

High confidence, non intronic somatic mutations were found by NGS in 35 patients, 32 of which had response data available. Gene expression and response data were available in 39 patients (Figure 1). For clarity, a detailed description of the most relevant variables evaluated in each patient in whom both response and molecular data were available (*n* = 46) is presented in Figure A1 along with the grade of response according to pTRG, NAR and mrTRG classifications. In the case of mrTRG, we grouped mrTRG1 and mrTRG2 in the “good” category and mrTRG4 and mrTRG5 in the “poor” category, with mrTRG3 considered “intermediate”. As shown, classification methods did not match exactly in classifying responses. Therefore, we established a dichotomic classification where good responders were those who achieved cCR and/or pTRG 0–1 (*n* = 18), while poor responders were those with pTRG 2–3 or tumors found unresectable during surgery without disease progression (*n* = 28). We termed this classification the Consensus Response. This was evaluated along with the other classifications (i.e., pTRG, NAR, mrTRG) in a principal component analysis (PCA) of whole gene expression data (Appendix A). The Consensus Response classification appears to better reflect the natural biological division between poor and good responders than other classifications. This led us to select it as our chosen method to define good and poor responders in our study.

### 2.2. Gene Expression Analysis and Validation

A total of 39 of the 50 patients had both microarray and Consensus Response data (Figure A1). As a first step, we applied the CMS classification [27], as the state of the art molecular classification of colorectal cancer. Most of these patients (35/39) were assigned to CMS2, the most common subtype among rectal tumors; the remaining four patients were CMS3 (Figure 2a). No association between CMS and response to nCRT was found. This may be related to the fact that the CMS subtypes were developed with 3962 colorectal samples, where only 450 samples were rectum tumors. Of note, none of the previously proposed predictive gene lists [19,20,21,22,24,25,28] were able to classify good and poor responders with reasonable specificity or sensitivity when applied to our cohort (Appendix A). Recently, the largest and most recent LARC cohort was published (Park et al [28]). Unfortunately, the small number of genes evaluated by the cohort prevented most of our inferential analyses to perform properly and could not be adequately evaluated. Nevertheless, we found significantly enriched pathways regarding interferon and immune response in good responders for this cohort as well (Appendix A).

To search for biological pathways that may distinguish Consensus Response good and poor responders, we performed GSEA-P analysis using curated gene sets. We observed an enrichment of cell cycle, DNA repair and metabolism pathways among the poor responders, and preponderant pathways of antigen presentation and immune cell activity, including interferon (IFN) 1 and 2 activity and B cell receptors, among the good responders (*p* < 0.05, Appendix A). A complementary pathway-centered analysis, PARADIGM, showed significant (*p* < 0.005) activation of genome maintenance and DNA repair pathways in poor responders (RAD51-BRCA2 homologous recombination complexes, FANCG) (Figure 2a). In contrast, terms related to B cell activity (BCR complexed with several effectors) and IFN-2 pathways (IFIT1 signaling) were upregulated in good responders. Furthermore, additional IFN-2-related terms (IRF1, IFIT3, IFN-gamma/JAK2 complexes), emerged when relaxing significant *p*-values to <0.05.

We conjointly used deconvolution analyses to explore the contribution of the tumor microenvironment and search for differences in cell populations. By using seven in silico cell population estimators we consistently found a significant enrichment of B cells in good responders (see ImSig in Figure 2b and Appendix A), irrespective of the method used. A small subset (*n* = 10) of these patients could also be tested by RNAseq, where the B cell terms also proved differential between good and poor responders (Appendix A), supporting this finding with another gene expression technique.

To validate the differential B cell content in good responders, we immunohistochemically assessed CD20 expression (Figure 3a) in the FFPE half of the pre-treatment tumor biopsies. Quantification of anti-CD20 positive staining by digital imaging analysis (Figure 3b) demonstrated that good responders had a higher percentage of CD20-positive staining areas within the tumor mass compared to poor responders (*p* = 0.047, Hedges’ g = −0.73); in comparison, no differences between good and poor responders were observed in the surrounding non-malignant tissue. As we noticed two distinct clusters in the good responder group of Figure 3b, we looked if there were associations to any of the clinical and pathological variables. None of the variables tested showed a clear pattern of association with any of these two clusters. An independent blinded analysis by two pathologists (Figure 3c) also showed a similar tendency towards finding lower B cell content in the tumor area of poor responders. Although not statistically significant because of the low number of samples, the differences seen rendered a Hedges’ g effect size of −0.45, which is considered a medium to large effect [29].

### 2.3. Mutational Profile

Out of the 42 patients with targeted sequencing data, 37 presented somatic mutations in exonic regions, splice sites or intronic boundaries of the evaluated gene panel. The remaining five patients (12%) showed no somatic mutations within the genes of GeneRead panel; these patients may have driver mutations in genes not evaluated by the panel, or their driver events could be copy-number aberrations or structural variations that were not analyzed in this study. Figure 4a summarizes findings of the 35 patients with non-intronic somatic mutations. Only two patients were assessed as dMMR (mismatch repair defficient) by immunohistochemical evaluation of mismatch-repair protein levels; consistently, these two patients had the highest rates of detected mutations. The remaining patients can be assigned to the non-hypermutated class of colorectal cancer.

Among the 37 patients with somatic mutations of any type, *APC* was the most frequently mutated gene, with 65% (24/37) of the patients showing one or two mutations, about half of them truncating, in the previously reported 1309–1450 hotspot region (Figure 4a,b, Appendix A). The next most frequently mutated gene was *TP53*, in 57% (21/37) of the patients, with point mutations mainly of the missense type, and one-third of the mutations located in well-known hotspots. Only one patient showed a likely pathogenic *BRAF* mutation, c.2128-3T>G, which affects a splice site. No *BRAF*${}^{V600E}$ was found in the analyzed patients. *ATM* and *PIK3CA* mutations were present in a quarter of the patients, again a frequency expected for colorectal cancer. Most *ATM* mutations were truncating nonsense/frameshift or damaging missense variants. On the other hand, four out of six patients presented known activating mutations in *PIK3CA*. The *RAS* family of genes was also frequently mutated, specifically *KRAS* in 48% (18/37) of patients and *NRAS* in 5% (3/37) (Figure 4a,b, and Appendix A). About half of the *KRAS* mutations affected codon 12 or 13 (Appendix A), as seen in other cohorts. *RAS* mutations were predominant in the poor nCRT responder group (16/24 patients, *p* = 0.023, Figure Figure 4b,c, left). Interestingly, patients with mutations in both *KRAS* or *NRAS* and *TP53* (KP-genotype) were only present in the poor responder group (12/24 patients, *p* = 0.001, Figure 4c, right).

### 2.4. Integration of Clinical and Multi-Omic Features for Response Prediction

We tested the association of nCRT with several dichotomic variables and potential confounding factors, including some already described in the literature [30]. As our cohort size limits statistical analysis, we explored a simple way of classifying patients binarily with quantitative variables. Initially we selected age and sex as confounding factors; blood parameters at diagnosis such as CEA levels and the neutrophil platelet score (NPS) [31]; variables evaluated by MRI before treatment such as the Extramural Vascular Invasion (EMVI), Circumferential Radial Margin (CRM), nodal status and lateral lymph node dissemination; immunohistochemical variables such as dMMR and CD20 staining for B cell evaluation, and molecular findings such as *RAS*, *TP53* and *ATM* mutational status, gene-expression derived CMS and scores. We selected the ImSig score as the most representative among the tools for cellular deconvolution analysis (Figure 2b). For ImSig scores, we defined values equal to or higher than the median ImSig score of the cohort as “high”, and the rest as “low”. We also constructed a DNA repair pathway score by selecting all significant PARADIGM scores for DNA repair-related terms (Figure 2a) and calculating a median for each patient. We termed this the median repair pathway score (MERPS). Patients were assigned as “high” or “low” depending on whether their MERPS were equal or higher or lower than the cohort median of MERPS, respectively (Figure 2c). Table 2 shows all dichotomic variables tested and their odds ratios (OR) for association with nCRT response. Low NPS, high ImSig scores and low MERPS proved to be significantly associated with good response, while the presence of *RAS* mutations and the co-occurrence of *RAS* and *TP53* mutations were significantly associated with poor response. Of note, while *ATM* mutations were not significantly associated with nCRT response, the probability of having a loss-of-function *ATM* mutation in good responders in our cohort was four times higher than in poor responders.

We then explored if these parameters also offer prognostic value. Given the relatively short follow-up time of this cohort, we limited our time-related analysis to RFS. Appendix A shows that a high ImSig B cell score was significantly associated with better RFS (*p* = 0.019). No other molecular variables were statistically significant in the RFS Kaplan-Meier analysis, although high MERPS and *RAS* mutant status suggested a slight tendency to be associated with poorer RFS. These variables were also included in Figure A1 in a patient per patient basis.

## 3. Discussion

Integrating multiple analytical approaches based on gene expression data, we identified a collection of curated signaling pathways and clinical features that are significantly associated with sensitivity or resistance to nCRT. We were able to infer positive regulation of genome maintenance, metabolism and cell cycle from poor responder transcriptomes, suggesting greater proliferation in these tumors. The good responder group displayed activated antigen presentation, inflammatory cytokines, and B cell-related signaling pathways; moreover, a higher density of B cells was detected in tumors of these patients. We infer from this that there are pre-existing biological characteristics related to quality response to nCRT, notably a less effective DNA replication and damage repair system and a more proficient B cell-mediated immune response within the tumor microenvironment.

Mutational analyses added a deeper layer to our transcriptomic findings. Considering that patients with constitutively activating *KRAS* mutations have worse prognosis across multiple tumor types including colorectal cancer [32], with worse overall survival in rectal cancer patients [33], the predominance of *KRAS* mutations among poor responders may reflect an intrinsic resistance to standard treatments. There is evidence of an association between *KRAS* mutations and response to conventional chemotherapy in colorectal cancer [34], although other studies have not reached significance, probably due to heterogeneity in stage selection and treatment [35]. The influence of *KRAS* mutations in response to nCRT in larger cohorts merits further investigation. We confirmed in our small series the previously reported strong effect of double *TP53* and *KRAS* mutations (KP) on nCRT response [23,36]. Since 100% of the KP mutants were poor responders, KP-mutant occurrence can be considered an indication of nCRT resistance.

The integration of potentially predictive features into our comprehensive analysis, lays the foundations for defining more accurate biomarkers of response to nCRT. We observed activation of DNA repair and genome maintenance pathways, concomitant with *RAS*/*RAS*-*TP53* mutations in the poor response patient group. DNA repair pathways, when abnormally suppressed by mutations, led to uncontrolled tumor growth; however, when the tumor is challenged by nCRT, a proficient DNA double-strand break repair system may help to manage the DNA damage associated with radiotherapy. It is known that DNA repair of double-strand breaks via homologous recombination is an effective mitigator of radio-induced damage [37]. Moreover, studies have detected that rectal tumors with a high percentage of Ku70-positive cells (the product of XRCC6, a DNA double-strand repair gene) tended to be radioresistant [38]. Interestingly, it is also known that p53 is implicated in suppressing excessive repair by homologous recombination to balance genomic stability, more specifically by modulating RAD51/52 and BRCA2 activity [39,40]. Thus, the enrichment of *TP53* loss-of function mutants among poor responders could be actively collaborating in the enhanced repair activity seen in poor responders by releasing the brake on RAD51/52 and BRCA2. In line with this, a recent publication showed that inhibition of the wild-type RAS pathway with targeted therapy downregulates both MMR and homologous recombination repair genes and enhances mutability [41]. This supports the hypothesis that a *RAS*-mutant tumor—in which this pathway is, on the contrary, constitutively activated—may also have a constitutively upregulated DNA repair pathway. Thus, a double *RAS*/*TP53* mutant may favor more effective repair of nCRT-induced damage. On the other hand, the tendency of good responders to harbor *ATM* loss-of-function mutations may be related to a relatively diminished capacity of repairing double-strand breaks, thus enhancing sensitivity to radiotherapy.

Our findings also suggest a role for B cells in the quality of the response to nCRT. Upregulation of immune response pathways related mostly to B cells and IFN (both 1 [i.e., alpha] and 2 [i.e., gamma]) in good responders may reflect the presence of a competent adaptive immune system ready to activate an effective antitumor response once the nCRT starts damaging tumor cells and liberating immune attractors such as damage-associated molecular patterns (DAMPs) and/or neoantigens [42]. Neighboring B cells may be activated by these antigens through B cell receptor interaction and act simultaneously by secreting inflammatory cytokines such as IFN and presenting neoantigens to CD4${}^{+}$ T cells in the context of their MHC class II ligands [43]. Likewise, we also observed significant enrichment of antigen presentation pathways in good responders across multiple analyses such as PARADIGM and GSEA. As nCRT treatment starts acting, this scenario might ease an appropriate antitumor immune response mediated by cross-priming of CD8${}^{+}$ T cells by MHC class-II-activated CD4${}^{+}$ T cells. A recent study indicated CD8${}^{+}$ tumor-infiltrating lymphocytes (TILs) are associated with a good nCRT response [18]. Of note, the higher relative values of circulating neutrophils in poor responders (reflected by higher NPS) and its association with poorer survival in the literature has been linked to the capacity of neutrophils to remodel the tumor microenvironment towards a more immune-resistant profile [44].

Recent studies proposed that radiation can induce reactions that resemble an antiviral response, supporting a key role of radiation in the stimulation of a tumor-specific immune response and tumor regression [45,46]. Radiotherapy-induced double-strand DNA breaks might trigger an IFN-1-mediated inflammatory response similar to that generated by the accumulation of viral DNA in the cytosol of infected cells, enhancing antigen presentation to effector T cells; this mechanism could be less efficient in a tumor with a more competent DNA repair pathway. This hypothesis links the two main findings of our study.

While we are aware of the sample size limitations of our hypothesis-eliciting work, validation of our findings with other technical approaches (i.e., CD20${}^{+}$ staining for B cells) supports that our results merit consideration. Other limitations such as tumor heterogeneity, tumor purity (in our case inferred from the adjacent FFPE tumor section) and differences between TNT and CRT treatments, were also overcome by this validation.

On the other hand, our study comprises one of the few series with current, state-of-the-art clinical and histological assessments. We also confirm previously reported deleterious clinical and molecular insights and propose for the first time characteristics associated with beneficial response, which may be combined in a rapid and cost-effective predictive evaluation. The impact of the significant variables described here as potential predictive biomarkers, as well as those which were not significant, merits further evaluation in larger numbers of well characterized LARC patients, including information about outcomes of neoadjuvant treatment. If confirmed, our findings should help to better identify potential poor responders and avoid unnecessary exposure to lengthy, morbidity-inducing and, most importantly, ineffective treatment.

## 4. Materials and Methods

This prospective translational study comprised the biological and molecular profiling of consecutive eligible LARC patients who underwent therapy at the Oncology Unit at Bonorino Udaondo Hospital (Buenos Aires, Argentina) from November 2015 to September 2018. Patients had to fulfil the following eligibility criteria: (a) available pre-treatment formalin-fixed paraffin-embedded (FFPE) biopsies, (b) histologically confirmed adenocarcinoma, and (c) completion of neoadjuvant therapy followed by TME surgery as the first therapeutic approach. Initial clinical staging was based on rectoscopy, thorax-abdomen computed tomography (CT) scan and pelvic magnetic resonance imaging (MRI). Clinical data collected from patients’ medical records were age at diagnosis, gender, distance to anal verge, risks factors according to ESMO rectal cancer guidelines [47], CEA and CA19.9 values, histological features (grade of differentiation and mucinous histology), mismatch repair (MMR) protein status by immunohistochemistry and neutrophil-platelet score (NPS). All patients signed the approved Informed Consent. All subjects gave their informed consent for inclusion before they participated in the study. The study was conducted in accordance with the Declaration of Helsinki, and the protocol was approved by the Udaondo Hospital Ethics Committee (Project Identification code HBU-ONCO-DEGENS, approved 18th May 2015) and the Instituto Leloir Institutional Review Board CBFIL (Project Identification code CBFIL#20, approved 30th May 2015).

Our standard routine approach for delivery of neoadjuvant therapy as the initial therapeutic approach define intermediate/locally advance rectal cancer as very low cT2-T3ab, cT3cd-T4, extramural vascular invasion (EMVI) positivity, high mesorectal nodes burden or mesorectal nodes unlikely amenable for quality TME, circumferential radial margin (CRM) involvement and lateral lymph node dissemination (LLND). All patients were assigned to standard pelvic long course radiotherapy (LCRT: 50.4 Gy in 28 fractions of three-dimensional conformal radiotherapy, 1.8 Gy per fraction, per day) with concurrent capecitabine (825 mg/m${}^{2}$/bid for 28 days), termed hereafter CRT. Patients with a high risk of systemic relapse (EMVI, high mesorectal node burden and LLND) underwent TNT, which comprises pre-treatment before the CRT with three cycles of CAPOX (130 mg/m${}^{2}$ of oxaliplatin on day 1 and capecitabine 1000 mg/m${}^{2}$/bid, days 1–14 every 3 weeks). Two cycles of capecitabine monotherapy (850 mg/m${}^{2}$/bid, days 1–14 every 3 weeks) was then administered until response assessment for all patients. Together, TNT and CRT are referred to as nCRT throughout this manuscript.

Response assessment was measured within 6-8 weeks of completing radiotherapy by digital rectal examination (DRE), CT and MRI (ymrTN and mrTRG [12,14]. Pathological tumor regression (pTRG) was evaluated on the surgical specimen using the Protocol for the Examination of Specimens from Patients with Primary Carcinoma of the Colon and Rectum, v.4.0.1.0 recommended by the College of American Pathologists (CAP) [11]. Response to nCRT was also evaluated using the NAR score [16]. Patients with low rectal tumors and clinical complete response (cCR) by DRE and diffusion-weighted MRI (DW-MRI) (ymrT0N0, mrTRG = 1, low/lack of signal in DW-MRI) were exempted from surgery and were followed up every three months for the first two years and every six months thereafter. The remaining patients underwent a TME 12 to 16 weeks after completing radiotherapy. Adjuvant treatment was considered for patients with postoperative residual tumor presence associated with histopathological high-risk factors. Results shown in this paper include follow up for progression/relapse and survival status until August 2019.

### 4.1. Sample Collection and Quality Control

All sample collection procedures were carried out according to institutional standard operating procedures for frozen and FFPE specimens based on international consortia recommendations. Baseline tumor biopsies were collected as part of the rectoscopy diagnostic procedure and were divided into two blocks: one block underwent snap-freezing with liquid nitrogen and the other was prepared as FFPE. The latter was analyzed for the presence of at least 60% adenocarcinoma with hematoxylin/eosin staining. The snap-frozen mirror biospecimen was processed for molecular studies, while the FFPE was stored for immunohistochemical studies. Cold ischemia times were strictly monitored and registered in order not to exceed 30 min from extraction to fixation (formalin or freezing). On the day of collection of the diagnosis tissue sample, peripheral blood samples were also collected according to standard operating procedures.

Total tumor DNA and RNA were extracted from each snap-frozen biopsy with Allprep DNA/RNA/miRNA Universal Kit (QIAGEN, Hilden, Germany) according to the manufacturer’s instructions. DNA and RNA were quality assessed and quantified using NanoDrop 2000 and Qubit Fluorometer’s High Sensitivity DNA Assay Kit or RNA Assay Kit, respectively (Thermo Fisher Scientific, Waltham MA, USA). RNA integrity was also evaluated for RIN > 6 on a 2100 Bioanalyzer (Agilent Technologies, Santa Clara CA, USA). Normal DNA was extracted from baseline peripheral blood using QIAamp DNA Mini Kit (QIAGEN, Hilden, Germany) and quality assessed and quantified as described above for tumor DNA.

### 4.2. Targeted Sequencing

We performed targeted sequencing of 38 genes associated with colorectal cancer using GeneRead DNAseq Colorectal Cancer Panel V2 (QIAGEN, Hilden, Germany). Tumor and normal DNA libraries were constructed and sequenced in an Ion Proton sequencer (Thermo Fisher Scientific, Waltham MA, USA). Adequately covered amplicons were defined as those having a minimum mean coverage depth of 2500x and 500x for tumor and normal samples respectively. The resulting read data were quality assessed using Ion Torrent Suite software version 5.8 and uploaded to the GeneGlobe Data Analysis Portal version 2.0 (QIAGEN, Hilden, Germany) for alignment to the human reference genome hg19 (GRCh37) and variant calling. Output VCFs were annotated using Cancer Genome Interpreter [48], Variant Effect Predictor [49] and PeCan Pie [50]. Annotation results were then merged, and variant filtering and prioritization was performed by an in-house protocol based on CAP guidelines [29] and manual curation according to Barnell et al [27]. The resulting high confidence somatic variants were analyzed using R package maftools [30] and compared with the TCGA rectal cancer dataset obtained from cBioPortal [51].

Additionally, one patient for whom NGS data from fresh frozen tissue were unavailable was tested for *KRAS*, *NRAS*, *BRAF* and *TP53* mutations in the FFPE sample as part of an external clinical-grade NGS testing service.

### 4.3. Gene Expression Analysis

Tumor RNA and a Universal Human Reference RNA (Stratagene, San Diego CA, USA) were amplified, differentially labeled and subsequently hybridized with Human Gene Expression v2 4x44K Microarrays using the Human Gene Expression v2 4x44K Microarray kit (Agilent Technologies, Santa Clara CA, USA). The output was analyzed with Feature Extraction 11.5.1.1 software (Agilent Technologies, Santa Clara CA, USA) and additional quality control and normalization were performed using the R package Agi4x44.2c [32].

The baseline transcriptomic profiles obtained were analyzed for Consensus Molecular Subtype (CMS) classification using the R package CMSclassifier [33], to determine pathway enrichment using GSEA-P (1.735 curated gene sets comprising all MSigDB Hallmark, KEGG and REACTOME) [34] and PARADIGM [35]. To estimate the variety of cell populations present in the tumor microenvironment, we used six independent in silico deconvolution tools: MCPclassifier [36], GSVA [37], xCell [38], TIMER [39], Imsig [40] and CIBERSORT [41] along with its improved implementation, MIXTURE [52]. Gene expression analyses results were plotted using MultiExperiment Viewer software [43].

### 4.4. CD20 Immunohistochemistry Validation

In silico deconvolution results regarding B cells were validated by anti-CD20 immunohistochemistry analyses. Double-blinded labeled microscopic slides were prepared from the FFPE tumor samples, processed in the BenchMark ULTRA IHC System, incubated with CONFIRM anti-CD20 L26 primary antibody (Ventana Medical Systems, Oro Valley AZ, USA) and ultraView Universal DAB Detection Kit as a secondary antibody (Ventana Medical Systems, Oro Valley AZ, USA).

Following whole slide scanning on the Aperio LV1 System (Leica Biosystems, Richmond IL, USA), tumor and normal areas were defined by two pathologists. The slides were then quantified by two separate methods: (1) An independent intratumoral and peritumoral CD20${}^{+}$ percentage estimation by two experienced pathologists, with the joint assessment of a third senior pathologist in cases of discrepancy; (2) Independent digital image analysis and quantification using immunohistochemistry profiler software [44]. Digitally analyzed tumor, normal, and total CD20${}^{+}$ percentages were calculated by dividing the CD20${}^{+}$ pixel count by the number of total pixels.

### 4.5. Statistical Analysis

Graph generation and statistical analyses were performed using available packages based on R software programming language (version 3.1.0) [53]. Differences according to response type in categorical variables such as mutational status for each gene or dichotomized MERPS score were compared using the Exact Fisher test, while differences in PARADIGM pathways between good and poor responders were assessed using the Student’s *t*-test. To test for differential immune cell populations from the in silico estimators, we used Mann-Whitney U test and calculated Hedges’ g to measure effect size with the R package DABEST [45]. Two-sided *p*-values < 0.05 were considered significant.

To check the potential prognostic value of the main predictive variables, we performed recurrence-free survival (RFS) analysis. RFS was defined as the interval from informed consent signature to the date of recurrence with medical confirmation. Recurrence-free patients were censored at the time of last follow-up. For patients known to have recurrence but missing an exact recurrence date, the date of the last medical examination was used. Kaplan-Meier analyses were performed using the R package Survminer [54].

### 4.6. Data Availability

The gene expression data reported in this study is available through GEO (ID: GSE150082), while DNA sequencing data can be found at SRA (ID: PRJNA633284).

## 5. Conclusions

Only 20-30% of locally advanced rectal cancer (LARC) patients achieve a complete response to pre-operative chemoradiotherapy (nCRT). Complete response remains the best surrogate of positive time-related treatment outcome. Advances to clarify the characteristics that determine response are needed for clinical decision-making. Our integrative molecular analysis in pretreatment biopsies showed enrichment of B cell-related genes and pathways in good responders, while positive regulation of DNA repair transcriptional programs and simultaneous *RAS* and *TP53* mutations were highly prevalent in poor responders. These original findings contribute to a better understanding of determinants of therapeutic outcomes for nCRT and represent a solid starting point for validating biomarkers which may spare rectal cancer patients who are potential poor responders from lengthy, ineffective, poorly tolerated and morbidity-inducing treatment.

## Figures and Tables

**Figure 1 cancers-12-02227-f001:**
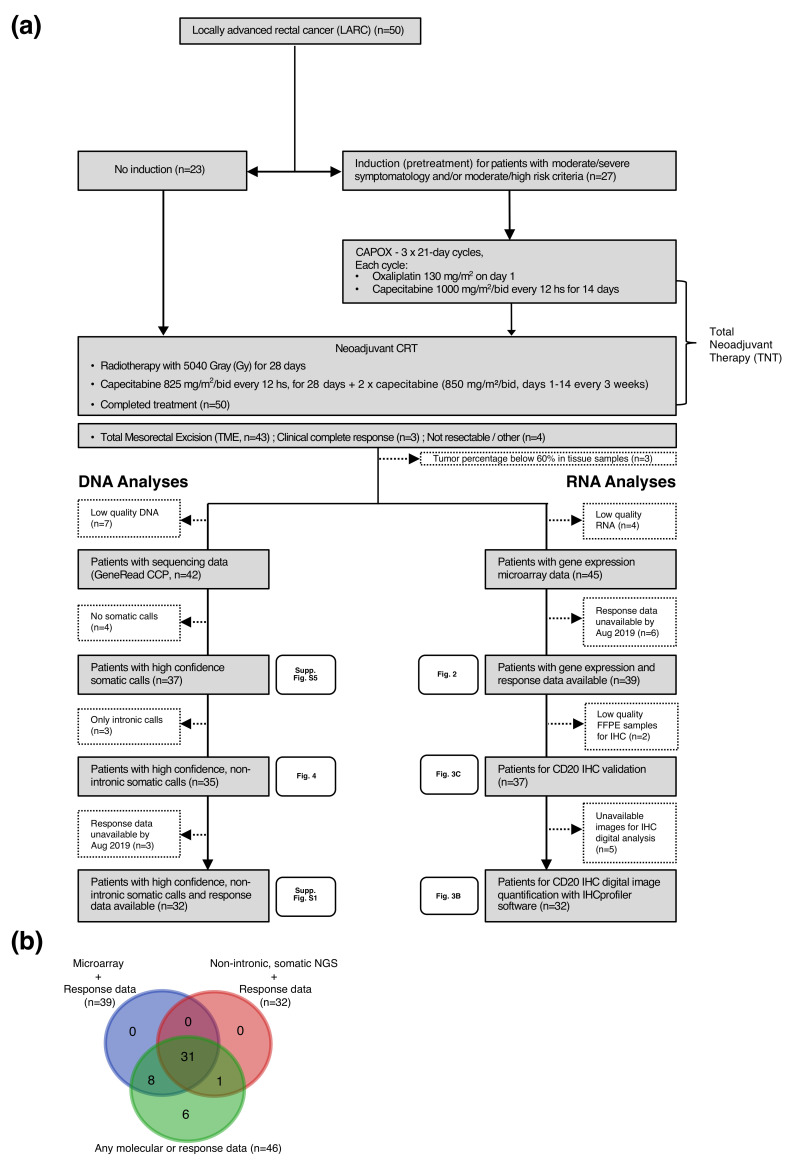
Description of the study design and participants. (**a**) Flow chart showing the cohort’s composition, outcomes and results. (**b**) Venn diagram displaying sample sizes and availability for the different molecular analyses performed.

**Figure 2 cancers-12-02227-f002:**
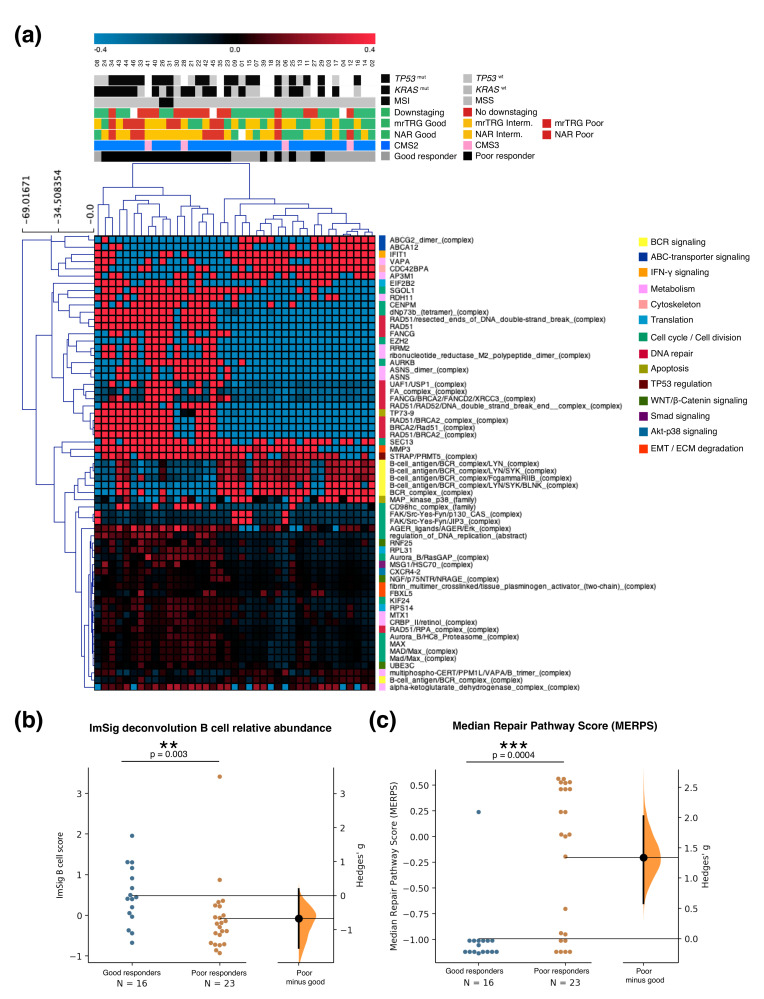
Differential gene expression analysis in good vs poor responders in the LARC cohort. (**a**) Heatmap of the 65 differential PARADIGM terms (*t*-test *p* < 0.005) between good and poor responders (columns), clustered by Pearson correlation. The upper panel depicts a category plot that summarizes the responder class of each patient according to the response evaluation methods (i.e., mrTRG, Consensus Response, NAR), as well as other relevant variables (downstaging, consensus molecular subtype [CMS] class, presence/absence of *TP53* and/or *KRAS* mutations, microsatellite instability [MSI] status). The red/blue bar of the top panel shows the range of median centered Integrated Pathway Activity scores from PARADIGM. Red terms correspond to higher pathway activation scores while blue represents lower scores. The lateral color code on the right of the heatmap panel summarizes the predominant biological significance of families of PARADIGM terms. (**b**,**c**) Gardner-Altman estimation plots showing the distribution of the abundance of B cells according to the ImSig algorithm (ImSig score), (**b**) and the distribution of the median of our PARADIGM-derived median repair pathway score (MERPS), (**c**) among good and poor responders. The significance of the differences of median (showed in the base of the histogram, with its corresponding 95% confidence interval) was tested by Mann-Whitney U test (** *p* < 0.01, *** *p* < 0.001). The right axis shows the Hedges’ g score, i.e., the estimated effect size, calculated as the differences between medians divided by the combined standard deviation, corrected for small sample sizes. Effect sizes over 0.8 correspond to large effects.

**Figure 3 cancers-12-02227-f003:**
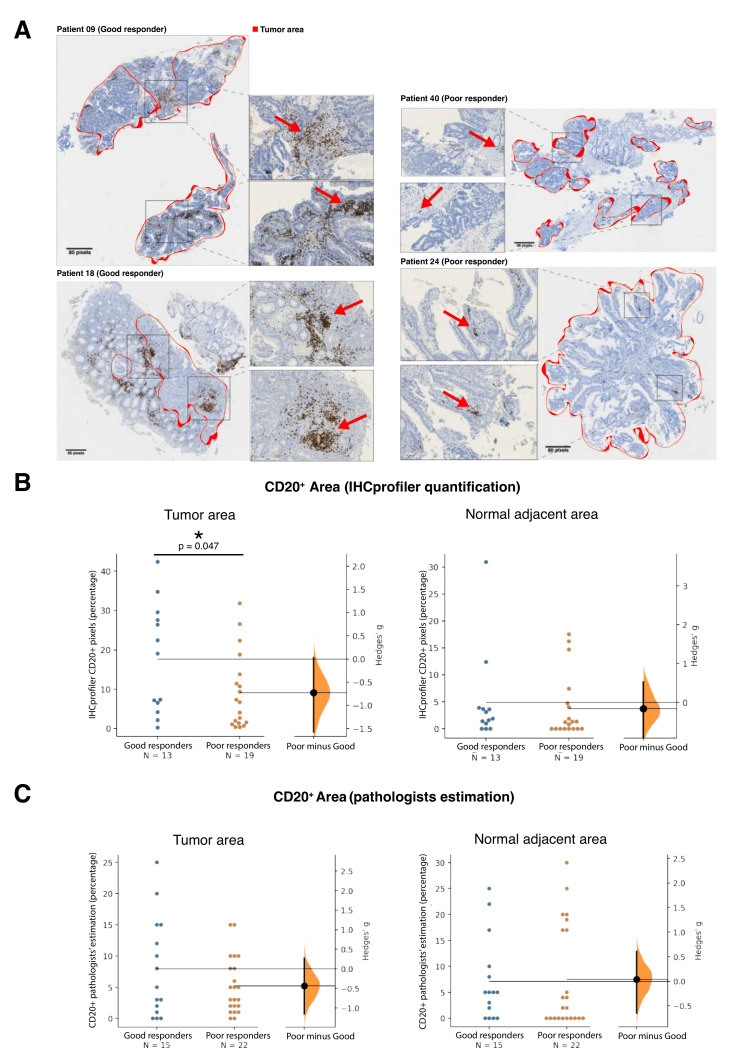
Validation of differential density of B cells in good and poor responders in the LARC cohort by anti-CD20 immunohistochemistry using slides obtained from the FFPE mirror image of the same pre-treatment tumor biopsies that were used for molecular analysis (see Methods). (**a**) H&E section images with CD20 staining for representative good (patients 09 and 18) and poor (patients 24 and 40) responders. Scale bars correspond to 80 pixels. Tumor areas are delimited with red lines, and the remaining areas are considered normal peritumoral tissue. Zoom sections of each scanned image correspond to a 10X magnification, with CD20 positive cells indicated with arrowheads. (**b**,**c**) Gardner-Altman estimation plots showing the distribution of the abundance of CD20 positive cells according to a digital quantification of stained areas (IHCprofiler; (**b**) and the pathologists’ evaluation (**c**), both in tumor and normal areas. The significance of the differences was tested by Mann-Whitney U test (* *p* < 0.05). Differences between (**b**) and (**c**) in the number of patients correspond to availability of FFPE samples and/or digital images.

**Figure 4 cancers-12-02227-f004:**
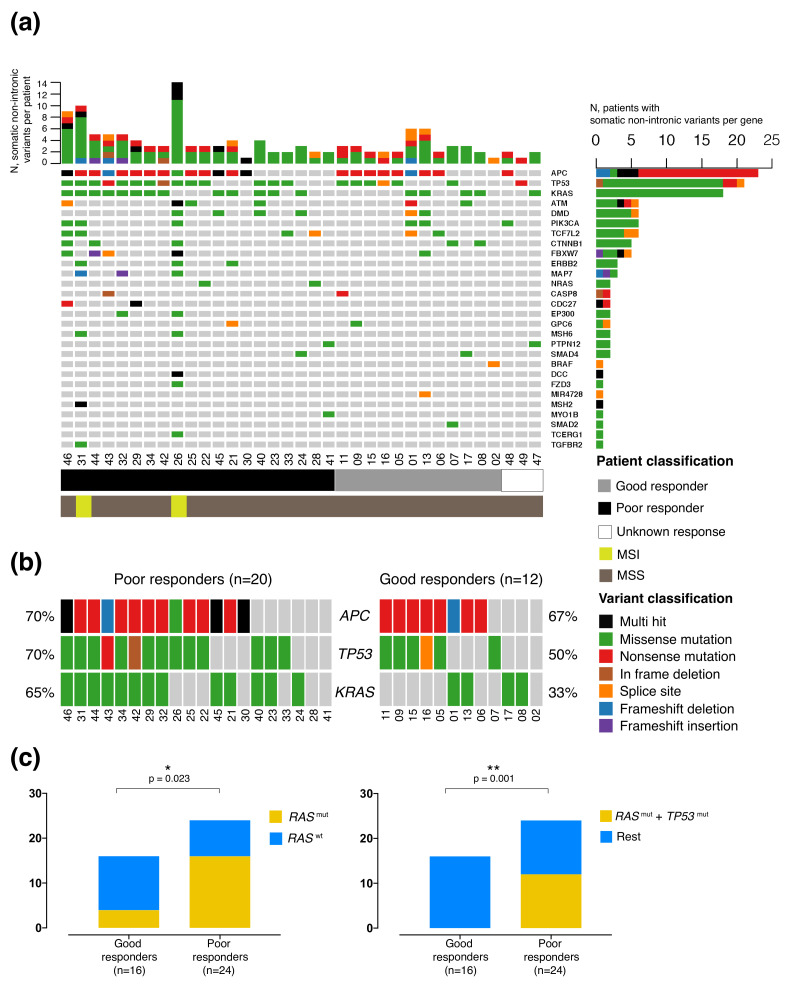
Spectrum of non-intronic somatic mutations found with the GeneRead Colorectal Cancer Panel in the LARC cohort (*n* = 35). (**a**) Waterfall plot depicting the type of somatic variants found per patient (columns) and per gene (rows). Intronic variants were not included. Colors indicate the type of mutation detected, as specified by the legend. Column bars at the top of the plot represent the number of non-intronic somatic variants present in each patient. Patients 26 and 31 were classified as having microsatellite instability (MSI). Color references at the bottom of the waterfall plot depict nCRT response according to the Consensus Response classification (CR) (black: poor response, grey: good response, white: off-treatment) and microsatellite status for each patient (brown: microsatellite stability [MSS], yellow: MSI). Bars to the right of the waterfall plot represent the number of patients who present somatic mutations in each evaluated gene. (**b**) Mutational status of the three most frequently mutated genes (*APC*, *TP53* and *KRAS*) in good and poor nCRT responders harboring non-intronic somatic mutations in any of those genes. (**c**) Contingency analysis of the presence of *RAS* mutations (left) or the co-occurrence of mutations in *RAS* and *TP53* (right) in good and poor responders. These plots and calculations include patients not evaluated by NGS panel but clinically tested for *RAS* /*TP53* mutations (See Appendix B for more details).The significance of the differences was tested by Fisher’s exact test (* *p* < 0.05, ** *p* < 0.01).

**Table 1 cancers-12-02227-t001:** A summary of clinical and pathological features of the LARC cohort.

Variable	*N* = 50
**Demographic and clinical variables**	
Age at diagnosis, median (range)	61 (34–75)
<50 years, *n* (%)	13 (26%)
Male, *n* (%)	37 (74%)
**Type of treatment, *n * (%)**	
nCRT	23 (46%)
TNT	27 (54%)
**Histological classification, *n*(%)**	
Adenocarcinoma, well differentiated	17 (34%)
Adenocarcinoma, moderately differentiated	27 (54%)
Adenocarcinoma, poorly differentiated	1 (2%)
Adenocarcinoma, differentiation data not available	5 (10%)
**Mucinous classification, *n*(%)**	
Mucinous	5 (10%)
Non-mucinous	45 (95%)
**Mismatch repair IHC assessment, *n* (%)**	
MMR proficient by IHC	47 (94%)
MMR deficient by IHC	3 (6%)
**Tumor localization, *n*(%)**	
Low rectum (<7 cm)	23 (46%)
Middle rectum (7–12 cm)	20 (40%)
Upper rectum (>12 cm)	7 (14%)
**Clinical staging by MRI, *n*(%)**	
T2-T3abN0, very low tumors, N0	3 (6%)
T3c-d/ T4, N0	18 (36%)
N1	19 (38%)
N2	10 (20%)
**MRI risk factors beyond TNM, *n*(%)**	
Initial EMVI ${}^{+}$	19 (38%)
Initial CRM ${}^{+}$	41 (82%)
Initial lateral lymph node dissemination	10 (20%)
**Post-nCRT outcomes, *n*(%)**	
Clinical complete response (‘Watch and wait‘ strategy)	3 (6%)
TME	43 (86%)
Not resectable during surgery	3 (6%)
Disease progression during treatment	1 (2%)

LARC = Locally Advanced Rectal Cancer; nCRT = Neodjuvant/preoperative Chemoradiotherapy; TNT = Total Neoadjuvant Therapy; MMR = Mismatch Repair; IHC = Immunohistochemistry; MRI = Magnetic Resonance Imaging; EMVI = Extramural Vascular Invasion; CRM = Circumferential Radial Margin; TME = Total Mesorectal Excision.

**Table 2 cancers-12-02227-t002:** Evaluation of clinical and molecular predictive factors in the LARC cohort using Consensus Response (CR) as a measure of response to nCRT.

Category	Dichotomic Value	Total	Good Response (CAP 0–1) *	Poor Response (CAP 2–3) ≠	*p*-Value (Fisher’s Test)	OR	95% CI
Age (years)	>50	35	15	20	0.487	2	0.434–7.793
	${}^{\leq }$ 50	11	3	8			
Sex	Male	33	15	18	0.197	2.778	0.673–10.54
	Female	13	3	10			
Treatment type	nCRT	22	10	12	0.295	0.6	0.182–1.979
	TNT	24	8	16			
CEA	>3 ng/mL	30	11	19	0.753	0.712	0.175–2.931
	${}^{\leq }$ 3 ng/mL	16	7	9			
**Neutrophil/Platelet** **Score (NPS)**	**high (1 or 2)**	12	**1**	**11**	**0.034**	**9.387**	**1.131–445.6**
	**low (0)**	34	**16**	**18**			
MRI-assessed initial extramural vascular invasion (EMVI)	Yes	17	6	11	0.761	0.773	0.237–2.454
	No	29	12	17			
MRI-assessed initial circumferential radial margin (CRM)	Positive	38	15	23	1	1.087	0.223–4.578
	Negative	8	3	5			
Nodal status	N0	22	9	13	1	1.154	0.382–3.514
	N1 or higher	24	9	15			
Lateral lymph node dissemination	Yes	9	2	7	0.447	0.4	0.077–1.854
	No	36	15	21			
Mismatch repair deficit	Yes	2	0	2	0.513	0	0–3.348
	No	44	18	26			
***RAS* mutation**	**Yes**	20	**4**	**16**	**0.023**	**0.167**	**0.049–0.663**
	**No**	20	**12**	**8**			
*TP53* mutation	Yes	22	7	15	0.335	0.467	0.138–1.653
	No	18	9	9			
*** RAS/TP53 * double mutation**	**Yes**	12	**0**	**12**	**0.001**	**0**	**0–0.282**
	**No**	28	**16**	**12**			
*ATM* mutation	Yes	8	5	3	0.235	3.03	0.576–12.79
	No	31	11	20			
Consensus Molecular Subtype (CMS)	2	35	14	21	1	0.674	0.044–10.31
	Other than 2	4	2	2			
% CD20${}^{+}$ in tumor (IHCProfiler)	high	16	8	8	0.473	2.145	0.422–11.957
	low	16	5	11			
% CD20${}^{+}$ in tumor (pathologists)	high	19	8	11	1	1.038	0.227–4.794
	low	17	7	10			
**B cell ImSig score (median)**	**High**	20	**12**	**8**	**0.022**	**5.625**	**1.394–19.18**
	**Low**	19	**4**	**15**			
**Median Repair** **Pathway Score (MERPS)**	**Low**	19	**13**	**6**	**0.001**	**12.28**	**2.548–47.65**
	**High**	20	**3**	**17**			

Variables with *p*-values 0.05 are highlighted in bold. Each variable was analyzed using the total number of subjects for which both the Consensus Response and the value of the corresponding variable were available. Contingency tables were analyzed by Fisher’s test. OR: odds ratio; CI: confidence interval for OR. ^*^ = CAP 0 also includes three patients with complete clinical response undergoing a ’watch and wait’ strategy (no surgery), categorized as good responders. ≠ = CAP 3 also includes three patients who were categorized as unresectable but free of metastasis at the moment of surgery, categorized as poor responders.

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
