# Peer review of "Pre-Existing Tumoral B Cell Infiltration and Impaired Genome Maintenance Correlate with Response to Chemoradiotherapy in Locally Advanced Rectal Cancer"

_cancers, 2020, doi:10.3390/cancers12082227_

Round 1

Reviewer 1 Report

This is an interesting, integrative analysis of molecular determinants of responses of locally advanced rectal cancer patients to neoadjuvant therapy. The authors employed targeted sequencing for somatic mutation analysis, and microarray analysis for transcriptome studies. The authors identify a co-occurence of RAS and p53 mutations to be associated with poor prognosis in this patient cohort, and support evidence for a prognostic potential of CD20+ cells.

Main criticism:

  1. The author tested several previously published transcriptome-based risk scores but could not validate these scores in their cohort. This points to the importance of clinical validation of risk scores or prognostic molecular markers. The authors should therefore interrogate other data sets regarding the co-occurence of RAS and p53 mutations and negative prognosis, e.g. in the TCGA data set. The reviewer realises that the TCGA mainly constitute of adjuvant cases, but a similar trend should be evident.  Other data sets may also be available.

Minor criticisms:

2. How were mucinous tumors defined?

3. The authors should mention that the CMS subtyping was originally performed predominantly on colon (not colorectal) cancers, and may have limited utility for rectal cancers.

4. Did the authors also try to build a random forest to include some of the clinical and pathological information in their risk score evaluation?

5. Table 2 should include a multivariate analysis.

6. Was there an association between type of p53 mutation and recurrence?

Author Response

Dear Reviewer 1,

Thank you very much for your valuable feedback. We believe your questions and suggestions will bring major improvements to our article. 

Please find our point-by-point response to your comments in the attached pdf.

We look forward to a positive evaluation for our manuscript for publication.

Yours sincerely,

Andrea S. Llera, Ph.D.

Reviewer 2 Report

Within this study, the authors performed multiple analysis for tumoral B cell infiltration and additional genomic analysis in patients with LARC, before and after nCRT. Although this topic is of broad interest, I have major concerns about the presented data:

  1. When 50 patients have been included in the study and pre-treatment tumor biopsies have been collected (according to Figure 1/Table 1), a drop out of 18 patients (Figure 4b) is too high for a good planning of this study?
  2. Table 1: TME is not a treatment response, but refers to a surgical technique. This has nothing to do with response. 8 patients could not be classified to adenocarcinoma. What happened to these patients in the following analysis? Excluded? This is important as mucinous cancers and adeno carcinoma have different biological outcomes and responses to nCRT.
  3. Figure 2 is contrary to Figure 1 for CD20 IHC. The authors missed to precisely explain which tissue has been used for which analysis.
  4. Figure 4b/c refers to normal adjacent tissue. Again, it is not clear which tissue has been used for analysis? The pretretment tissue according to Figure 1 or the tumor specimen after surgery according to Figure 2? This is extremely important to clarify as chemoradiation itself induces inflammation in the tumor and adjacent tissue and has major impact on the analysis of B cell infiltrates. A better way of showing the importance of B cell infiltrates and B cell signatures would be to compare the pre-treatment tissue biopsies with the tissue specimen after nCRT and surgery. This would give the reader a better understanding if B cell infiltrates in the pre-treatment tumor tissue have in fact an effect on the response to nCRT. Here, the authors miss to make a precise statement easy to follow by the reader.
  5. Figure 4b shows two cohorts in the tumor area (7 above 20, and 6 below 20). It would be interesting to correlate these patients with the pathological subtype. 
  6. The data from the most reliable source for the effect of B cell infiltrates, 2 independent pathologists (Figure 4c/Table 2), did not show significance, thus the conclusion of the manuscript is not supported by the most important data. Additionally, only 32 patients have been assessed in this analysis which makes it even harder to being convinced from the presented data.

Minor:

  1. There are several disparities in the figures and text (Figure 2 and text mentions supplementary figures that are not present, probably from an earlier submission).
  2. What about the effects on T cell markers? CD3/CD4/CD8? Thiscan be used as controls to emphasise the importance of the presumably discovered B cell phenotype in good responders.
  3. It would make much more sense to validate all used techniques in the same patient cohort. This would strengthen the paper. However, the minimum number of patients used in one analysis was 32, which might be not enough to reach statistical significance.

Author Response

Dear Reviewer 2,

Thank you very much for your valuable feedback. We believe your questions and suggestions will bring major improvements to our article.

Please find our point-by-point response to your comments in the attached pdf.

We look forward to a positive evaluation for our manuscript for publication.

Yours sincerely,

Andrea S. Llera, Ph.D.

Reviewer 3 Report

Congratulations for your great work.

Authors showed how to predict tumor response to nCRT using comprehensive molecular approach.

The result of this study would not influence on real world practice, but, I believe it would be a basic for further investigation.

Although cohort size was not sufficient, I understand it is hard to obtain or collect specimen for these specific cohort.

One thing that I`d like to mention is inclusion. Authors included both patietns who received nCRT and induction chemotherapy. Induction chemotherapy possibly effect on tumor response which  is not radioresponse but chemo-response.

I cannot find seperate analysis for these 2 group of patients. I recommend to do analyze respectivley.

It is a reliabel method to ascertain the results of this  study is develop biomarker for tumor response to radiotherapy.

Author Response

Dear Reviewer 3,

Thank you very much for your valuable feedback which will bring major improvements to our article. We really appreciate your understanding regarding our research results and caveats.

Please find our point-by-point response to your comments in the attached pdf.

We look forward to a positive evaluation for our manuscript for publication.

Yours sincerely,

Andrea S. Llera, Ph.D.

Round 2

Reviewer 1 Report

The authors have submitted a revised MS. Unfortunately several issues raised have not been addressed, the authors claim low statistical power for not addressing eg Random Forest approaches or a sub-analysis of p53 mutation type, which is understandable.

However the authors should investigate p53/KRAS double mutant responses (ie recurrence) in the adjuvant setting in rectal cancer/CRC from the TCGA.  Indeed, responses to neoadjuvant therapy is an excellent predictor of long term outcome/recurrence in rectal cancer. It would greatly strengthen the paper if such an analysis would be added. 

Reviewer 2 Report

The authors provided additional explanation and data to answer my previous comments. The authors state, that the analysed specimens correspond entirely to pre-treatment rectal tumor biopsies. Please clarify in the Material and Methods section or in the legends of figure 4 what the term "normal adjacent tissue" reflects to. Is it (i) the "non-tumor cell infiltrated area" of the tumor biopsy; or (ii) the "normal tissue" from an independent biopsy couple inches away from the tumor (which would be unusual as the performing physician normally only biopsies the tumor for diagnostics). This is important for the reader to know as it has impact on the amount of infiltrating leukocytes (and thus CD20+ B cells) in the analysed area.